

# Land suitability assessment for wheat-barley cultivation in a semi-arid region of Eastern Anatolia in Turkey

Bulut Sarğın and Siyami Karaca

Faculty of Agriculture, Department of Soil Science and Plant Nutrition, Van Yuzuncu Yil University, Van, Turkey

## ABSTRACT

The efficient use and sustainability of agricultural lands depend heavily on the characteristics of soil resources in a given area, as different soil properties can significantly impact crop growth and yield. Therefore, land suitability studies play a crucial role in determining the appropriate crops for a given area and ensuring sustainable agricultural practices. This study, conducted in Tusba District-Van, Turkey, represents a significant advancement in land suitability studies for wheat-barley cultivation. Using geographic information systems, the analytical hierarchical process method, and the standard scoring function, lands were determined based on the examined criteria for the suitability of wheat-barley cultivation. One of this study's main findings is identifying critical factors that influence the suitability of land for wheat-barley cultivation. These factors include slope, organic matter content, available water capacity, soil depth, cation exchange capacity, pH level, and clay content. It is important to note that slope is the most influential factor, followed by organic matter content and available water capacity. A Soil Quality Index map was produced, and the suitability of wheat-barley production in the studied area was demonstrated. More than 28% of the study area was very suitable for wheat-barley production (S2), and more than was 39% moderately suitable (S3). A positive regression ($R^2 = 0.67$) was found between soil quality index values and crop yield. The relationship between soil quality index values and crop yield is above acceptable limits. Land suitability assessment can minimize labor and cost losses in the planning and implementation of sustainable ecological and economic agriculture. Furthermore, land suitability classes play an active role in the selection of the product pattern of the area by presenting a spatial decision support system.

## INTRODUCTION

In order to meet the increasing food demand with the rising world population, agricultural production should be brought to a level that meets these needs. Effective and appropriate use of agricultural lands is essential in protecting soil and water resources. Excessive use of soil and water resources causes significant damage to the ecological system. In particular, the decrease in the soil quality of agricultural lands leads to reduced productivity and plant diversity (*Jin et al., 2019*; *Dereumeaux et al., 2020*; *Jiang et al., 2020*; *Viana et al., 2022*). Soil degradation, the main element of the ecological system, threatens sustainable

Corresponding author
Bulut Sarğın, bulutsargin@yyu.edu.tr

land use. Sustainable agriculture is defined as the rational and effective use of land resources. For this reason, it is necessary to prepare regional plans and product patterns by considering agricultural policies and environmental concerns to ensure the sustainability of agricultural lands. Land suitability assessment is a scientific planning and management model to establish an ecological-economic balance (*Musakwa, 2018*; *Mihoub et al., 2022*; *Mohammed et al., 2022*). This model guides decision makers and producers by evaluating the suitability of the plant pattern to be grown in the field. Land suitability assessment plays an important role in determining the most appropriate use of lands and creating a sustainable environment for agricultural production. Determining the most appropriate use of the land is aimed for the farmers to produce efficiently and protect natural resources (*Talukdar et al., 2022*). Scientific analysis, classification, and use plans of soils provide the necessary data for the protection of agricultural areas. It plays an important role in the planning of investments such as soil conservation, irrigation, land use planning, and consolidation by determining the use potential of the soils. Determining the qualities of soil and water resources by evaluating them with participatory methods is used in the development of socio-economic, environmental efficiency, and protection methods (*Ennaji et al., 2018*; *Shakya et al., 2021*). Qualitative analysis of environmental factors such as soil properties, topography, and climate and quantitative evaluations based on yield estimates are important criteria in land suitability assessment studies. With these studies, the physical, chemical, and biological properties of the soil are evaluated together, and the soil quality index and suitability of the soil are determined. When parameters are evaluated alone, it is impossible to obtain sufficient information about the extent to which soil quality is affected (*Askari & Holden, 2015*; *Malone et al., 2015*; *Dengiz & Özyazıcı, 2016*; *Budak et al., 2017*; *Shakya et al., 2021*). The analytical hierarchy process (AHP) method has gained importance in determining soil quality in recent years. In addition, GIS-based multi-criteria decision-making methods (MCDM) using the AHP approach are successfully applied in determining land suitability in different soil and land conditions (*Dengiz & Sarıoğlu, 2013*; *Dağlı& Çağlayan, 2016*; *Mihoub et al., 2022*). In agricultural suitability analysis, it is possible to analyze and visualize environmental and spatial geographical data accurately and up-to-date by integrating geographic information systems (GIS) with the MCDM approach. With these applications, it is possible to produce objective and consistent information through multi-factor criteria. Considering the current situation, it is essential to study the efficient and effective use of limited agricultural lands to ensure sustainable economic growth and development of countries. The study was carried out in the Alaköy and Atmaca areas of Tuşba-Van district in Turkey. The reason for choosing the region is that there are no similar studies, and organic-sustainable agricultural practices are followed. The wheat-barley growing system is the dominant crop pattern; therefore, it was chosen in the scope of the current research. Suitable areas for ecological-economic cultivation were determined for wheat-barley production using the integration of the AHP approach with the GIS technique. The relationship between productivity and economy was evaluated by creating a model. It is thought that labor and economic losses will be prevented when the producers use the lands according to the determined suitability classes. In addition, similar

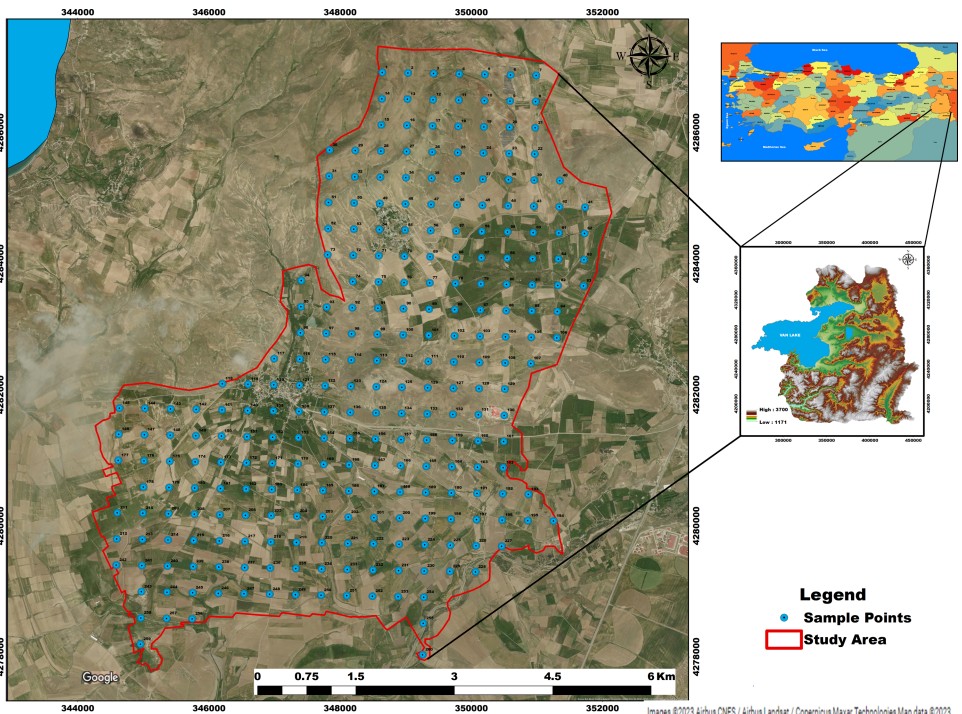

**Figure 1   Location of study area.** Satellite image credit: https://www.google.com/maps.

studies to be carried out in different locations and with crop patterns will contribute to sustainable agriculture without damaging ecological resources.

## MATERIALS AND METHODS

### Location

The study was carried out in an area of approximately 4207 ha in the Alaköy and Atmaca districts of Tusba-Van province, which is included in the 1:25,000 scaled K50c1 and K50d2 sheeted topographic maps. The study area is located between the coordinates 4278,000–4288,000 K and 344,000–352,000 D (Zone 38-UTM, m) (Fig. 1).

According to the long-term (1980–2020) climate data of the Van Regional meteorology station, the annual average temperature is 9.8 °C, and the annual average precipitation is 406.8 mm. The highest precipitation is in April with, 54.5 mm, and the lowest precipitation is in July, with 6.8 mm. The study area is ''Mesic'', and the soil moisture regime is ''Xeric'', according to *Soil Survey Staff (1999)*. The research area includes sandstone, claystone, marble, pebble, and limestone, formed in different geological periods (Cretaceous-Jurassic, Lower Cretaceous, Upper Cretaceous, Pliocene, Upper Paleozoic, Quaternary and Oligocene -Lower Miocene).

### Methodology

The topographic structure in the study area does not contain similar features, showing differences even in a short distance. Sampling was done within the framework of

**Table 1  The methods applied for the analysis of the physical, chemical and fertility properties of the soil.**

| Parameters | Unit | Method | References |
|---|---|---|---|
| Texture (sand, clay and silt) | % | Hydrometer method | *Bouyoucos (1951)* |
| Hydraulic Conductivity (HC) | (cm h$^{-1}$) | Darcy method | *Özdemir (1998)* |
| Volume Weight (BD) | (g cm$^{-3}$) | Intact Sample | *Blake & Hartge (1986)* |
| Useful Water Capacity (AWC) | (%) | Calculated from taking difference between FC and PWP | *Klute (1986)* |
| Soil reaction (pH) | 1:2.5 | Soil-Water Suspension | *Bayraklı (1987)* |
| Electrical Conductivity (EC) | dS m$^{-1}$ | Soil-Water Suspension | *Richards (1954)* |
| Organic matter (OM) | % | Walkley-Black method | *Jackson (1958)* |
| Lime (CaCO$_3$) | % | Calcimeter method | *Soil Survey Staff (1993)* |
| Available phosphorus | mg kg$^{-1}$ | Olsen method | *Olsen (1954)* |
| Total Nitrogen | % | Kjeldahl methods | *Kacar (1994)* |
| Exchangeable Cations (Ca$^{++}$, Mg$^{++}$, Na$^+$, K$^+$) | cmol kg$^{-1}$ | Ammonium acetate method | *Rhoades (1986)* |
| CEC (Cation Exchange Capacity) | cmol kg$^{-1}$ | Sodium acetate method | *Rhoades (1986)* |
| Microelements (Fe, Cu, Zn, Mn) | mg kg$^{-1}$ | DTPA extraction method | *Lindsay & Norvell (1978)* |

geostatistical methods to determine the point values and spatial distributions of the parameters of soil properties in a total area of 4,207 ha. Surface soil (0–30 cm) samples of 260 points were collected in the study area by making 400 × 400 m gridding in order to determine the soil properties. Degraded and undisturbed soil samples were taken from the selected points. GPS was used to identify the coordinates of each sampling point.

This study determined the importance levels of soil parameters for wheat-barley production. Physical (slope, depth, sand, clay, silt, hydraulic conductivity, bulk density, and available water capacity), chemical (pH, electrical conductivity, lime, cation exchange capacity, exchangeable sodium percentage, and organic matter), and productivity properties (nitrogen, phosphorus, available potassium, calcium, magnesium, sodium, manganese, iron, copper, and zinc) of 24 soil parameters were examined. These features were divided into subgroups according to their importance for wheat-barley production with expert opinions and literature support (*Dedeoglu & Dengiz, 2019*; *Günal et al., 2022*; *Kılıç et al., 2022*). The methods used to determine the physical, chemical, and fertility properties of soil samples are given in Table 1.

The soil data (physical, chemical, and productivity characteristics) and data processing flow chart used to determine the land suitability and obtain the maps are given in Fig. 2.

Spatial distribution maps of soil properties were prepared using the most common interpolation methods. The methods used are inverse distance weighting (IDW), the radial based function (RBF), and Kriging methods. While methods 1, 2, and 3 are used in the IDW method, the thin plate spline (TPS), completely regularized spline (CRS), and spline with tension (ST) methods are preferred in the RBF method. In the Kriging method, Natural (Ordinary), Simple (Simple), and Universal (Universal) methods are used. The RMSE approach was chosen, and 15 methods were compared while generating interpolation distribution maps. Among these methods, the method that gives the lowest square root (RMSE) of the mean square error was chosen as the most appropriate method.

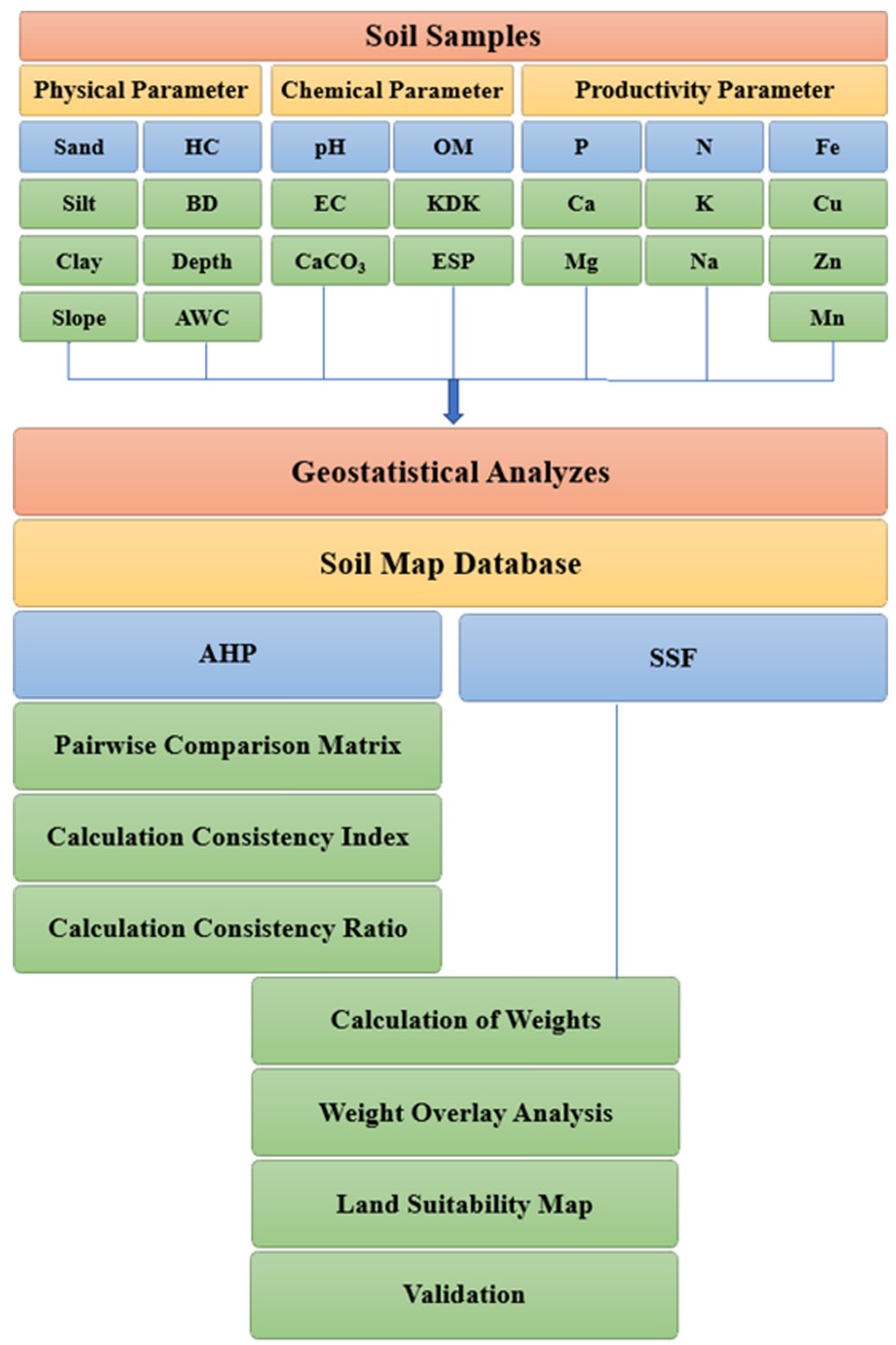

**Figure 2** Data processing flow chart for examining land suitability.

Equation (1) was used to calculate the RMSE value.

$$\text{RMSE} = \sqrt{\frac{\sum (Z_{i^\star} - Z_i)^2}{n}} \tag{1}$$

$Z_i$ = estimated value, $Z_i^\star$ = measured value and $n$ = number of samples.

## Land suitability parameters and soil requirements for wheat-barley cultivation

Wheat can be grown in many soils ranging from sandy loam to clayey texture. The water and nutrient holding capacity of sandy-loamy bodies is less than that of clayey-loamy soils. For this reason, soils with clay loam texture are more accepted. Wheat can be grown in soils with pH values in the range of 5.2–8.5, while different researchers have reported the optimum pH range value of 6.0–8.2 (*Mandal et al., 2020*; *Pilevar et al., 2020*; *Mihoub et al., 2022*; *Seyedmohammad & Navidi, 2022*). Wheat is less tolerant of salinity (>4 ds m$^{-1}$) during germination. When the EC values of the soils increase, wheat yield losses are 10% at 7.4 dS m$^{-1}$, 25% at 9.5 dS m$^{-1}$, 50% at 13 dS m$^{-1}$, and 100% at 20 dS m$^{-1}$, respectively, as reported by researchers (*Brady & Weil, 2008*; *Seyedmohammad & Navidi, 2022*).

Researchers reported that the average nutrient content of wheat soils would be sufficient for the requirements (*Sys Ir et al., 1993*; *El Baroudy, 2016*; *Karimi et al., 2018*). It is desired that the effective soil depth accepted in wheat cultivation should be between 120 and 150 cm. Under optimum conditions, a soil depth of less than 90 cm limits plant root growth (*Fischer et al., 2021*; *Navidi et al., 2022*). The fact that the exchangeable sodium percentage of the soil is less than 15% is very important in terms of plant growth and soil quality index. If the exchangeable sodium percentage is high, the soil pH will increase, and the nutrients in the soil will turn into useless forms (*Boyer et al., 2008*; *Pilevar et al., 2020*). In terms of other soil components, the threshold values of the reference values determined for wheat cultivation have been reported by different researchers. The desired limit range values of the 22 reported soil parameters are given in Table 2. Barley cultivation is not sensitive compared to wheat. It is cultivated in more barren soils. Barley can be grown in any soil conditions where wheat is grown. Therefore, our study's methodology was created based on the wheat plant, which has effective breeding criteria. Effective selection criteria were determined over the soil factors demanded by wheat.

## Scoring of soil parameters and weighting with AHP

Properties related to land and soil quality indices in terms of wheat-barley farming were scored with values between 0 and 1. Standard scoring functions (SSF) given in Table 3, which are widely used in the literature, were used for the scoring process. If a parameter achieves a high score value, SSF indicates a positive relationship (more is better) between soil quality and this parameter, so positive SSF is used. In other cases, negative SSF is used to obtain the desired parameters in good soil quality (less is better) (*Wymore, 1993*; *Karlen & Stott, 1994*; *Masto et al., 2008*; *Liu et al., 2018*).

Positive functions for depth, AWC, HC, clay, CEC, OM, N, P, Ca, Mg, Fe, Cu, Zn, and Mn were used as soil properties. In addition, a negative scoring function was preferred for

**Table 2  Land suitability values of parameter classes for wheat farming.**

| Suitability class* | S1 | S2 | S3 | N1 | N2 | |
|---|---|---|---|---|---|---|
| **Physical soil characteristics** | **100–80** | **79–60** | **59–50** | **49–20** | **19–0** | **Referances** |
| Texture | CL, SiCL, SCL, C < 60 s | SC, L, C < 60 | SCL, C > 60 | SL, Lfs | Cm, SiCm, LcS, fS | |
| Depth (cm) | >120 | 120–90 | 90–50 | 50–20 | 20–0 | *FAO, 1976* |
| Slope (%) | 0–6 | 6–12 | 12–20 | 20–30 | >30 | *Sys Ir et al., 1993* |
| BD (g cm$^{-3}$) | 1.0–1.3 | 1.3–1.4 | 1.4–1.6 | 1.6–1.8 | | *El Baroudy., 2016* |
| HC (cm h$^{-1}$) | <0.5 | 0.5–2.0 | 2–6.25 | >6.25 | – | *Mihoub et al., 2022* |
| AWC (%) | >50 | 50–20 | 20–15 | <15 | – | |
| **Chemical soil characteristics** | **100–80** | **79–60** | **59–50** | **49–20** | **19–0** | **Referances** |
| pH | 5.5–7.0 | 7.0–7.8 | 7.8–8.5 | >8.5 | – | |
| EC (dS m$^{-1}$) | 0–4 | 4–6 | 6–8 | 8–12 | >12 | *El Baroudy., 2016* |
| CaCO$_3$ (%) | <30 | 30–40 | 40–50 | 50–60 | >60 | *Karimi et al., 2018* |
| OM (%) | >2 | 1–2 | 1–0.5 | <0.5 | | *Mihoub et al., 2022* |
| CEC (cmol kg$^{-1}$) | >16 | 8–16 | <8 | – | – | *Fischer et al., 2021* |
| ESP (%) | <10 | 10–15 | 15–20 | >20 | | |
| **Soil fertility characteristic** | **100–80** | **79–60** | **59–50** | **49–20** | **19–0** | **Referances** |
| N (%) | >0.320 | 3.2–1.7 | 1.7–0.9 | 0.9–0.45 | <0.45 | |
| P (mg kg$^{-1}$) | >80 | 80–25 | 25–8 | 8–2.5 | <2.5 | |
| K (cmol kg$^{-1}$) | >2.56 | 2.56–0.74 | 0.74–0.28 | 0.28–0.13 | <0.13 | |
| Ca (cmol kg$^{-1}$) | >50 | 50–17.50 | 17.50–5.75 | 5.75–1.19 | <1.19 | *FAO, 1976* |
| Mg (cmol kg$^{-1}$) | >12.50 | 12.50–4.00 | 4.00–1.33 | 1.33–0.42 | <0.42 | *Sys Ir et al., 1993* |
| Na (cmol kg$^{-1}$) | 0–0.20 | 0.21–0.30 | 0.31–0.70 | 0.71–2.0 | >2.0 | *Sillanpaa, 1990* |
| Mn (mg kg$^{-1}$) | >170 | 50–170 | 14–50 | 4–14 | <4 | *El Baroudy., 2016* |
| Cu (mg kg$^{-1}$) | >0.2 | – | – | – | <0.2 | *Fischer et al., 2021* |
| Fe (mg kg$^{-1}$) | 2–4.5 | 1–2 | 1–0.2 | >4.5 | <0.2 | |
| Zn (mg kg$^{-1}$) | >1 | 1–0.5 | 0.5–0.25 | <0.25 | | |

slope, BD, sand, silt, pH, EC, CaCO$_3$, Na, and ESP. Field suitability analysis of wheat-barley was carried out by evaluating multiple criteria. Land suitability assessment was made using Linear Combination Technique by weighting the criteria that affect the agricultural land use of plants according to the order of importance. Each criterion was divided into sub-criteria and scored by giving a numerical value. The sub-criterion scores were multiplied by the weight value of the criteria they belong to, and the criteria were combined on the same scale. Equation (2) was used to calculate the most suitable lands for wheat-barley cultivation.

$$(SQI) = \sum_{i=1}^{n} WixXi \tag{2}$$

Here, SQI: total land suitability score for wheat-barley cultivation, Wi: weight value of parameter i (weights with AHP), Xi: sub-criteria score of parameter i (from scoring), n: total number of parameters considered. The values calculated using the Linear Combination Technique for each mapping unit in the soil database were classified according to Table 4. A land suitability map was created for wheat-barley crops by considering this classification, soil, and land quality parameters (*Gugino et al., 2009*). Multi-Criteria Decision Making

**Table 3  Standard scoring functions used according to soil properties.**

| Parameters | FT | SSF |
|---|---|---|
| Slope (%) | N | |
| BD (g cm$^{-3}$) | N | |
| Sand (%) | N | |
| Silt (%) | N | |
| pH | N | |
| EC (dSm$^{-1}$) | N | |
| CaCO$_3$ (%) | N | |
| Na (cmol kg$^{-1}$) | N | |
| ESP (%) | N | |
| Depth (cm) | P | |
| AWC (%) | P | |
| HC (cm h$^{-1}$) | P | |
| Clay (%) | P | |
| CEC (cmol kg$^{-1}$) | P | |
| OM (%) | P | |
| N (%) | P | |
| P (mg kg$^{-1}$) | P | |
| Ca (cmol kg$^{-1}$) | P | |
| Mg (cmol kg$^{-1}$) | P | |
| K (cmol kg$^{-1}$) | P | |
| Fe (mg kg$^{-1}$) | P | |
| Cu (mg kg$^{-1}$) | P | |
| Zn (mg kg$^{-1}$) | P | |
| Mn (mg kg$^{-1}$) | P | |

$$f(x) = \begin{cases} 0.1 & x \leq L \\ \{1-0.9\}x\dfrac{x-L}{U-L}+0.1 & L \leq x \leq U \\ 1 & x \geq U \end{cases}$$

$$f(x) = \begin{cases} 0.1 & x \leq L \\ \{0.9\}x\dfrac{x-L}{U-L}+0.1 & L \leq x \leq U \\ 1 & x \geq U \end{cases}$$

(MCDM) methods are used in issues involving multiple quality and quantity variables (*Timor, 2011*). AHP is an MCDM method introduced by *Saaty (2008)*, and it creates a hierarchy among variables depending on the expert's experience and the study's purpose. In this way, quantitative weights reflecting the importance of each variable are determined (*Srdjevic et al., 2010*). Therefore, AHP is a method used in studies where many qualitative and quantitative variables are evaluated together (*Panchal & Shrivastava, 2022*). The AHP method is based on a pairwise comparison matrix where all criteria are compared with each other (*Saaty, 2008*). The importance values of the variables that make up the matrix used in the study are given in Table 5.

After the sub-parameters are scored and created, the values of the basic parameters emerge. Revealing the fitness map depends on the main parameter values. Each parameter has different effect levels. In order to achieve this effect, geographical parameters should be compared with each other. The comparison of the parameters was carried out according to *Saaty (1980)* (Table 5). In binary comparison, two parameters are compared with each other, and the comparison depends on the judgment of the decision maker (*Öztürk & Batuk, 2010*). Pairwise comparison matrix (n) criteria is made with [n(n-1)/2] equation

**Table 4  Soil quality index classes for wheat and barley.**

| Class | Identification | Index value | Class |
|-------|---------------|-------------|-------|
| I | Very good | 80–100 | S1 |
| III | Good | 60–79 | S2 |
| III | Medium | 50–59 | S3 |
| IV | Weak | 20–50 | N1 |
| V | Bad | 0–19 | N2 |

**Table 5  Scale used for pairwise comparison of preferences in the AHP technique (*Saaty, 1980*).**

| Verbal preference provision | Explanation | Value |
|---------------------------|-------------|-------|
| Equal preference | Two activities contribute equally to the goal | 1 |
| Partially preferred | Experience and judgment partially favor one activity over another | 3 |
| Highly preferred | Experience and judgment highly favor one activity over another | 5 |
| Strongly preferred | It is strongly preferred over an activity value and its dominance is easily seen in practice. | 7 |
| Definitely not preferred | Evidence for choosing an activity over its value is highly reliable. | 9 |
| Intermediate values | Values falling between two consecutive jurisdictions to be used when compromise is required | 2, 4, 6, 8 |

**Table 6  Random index values used in calculating the consistency ratio and varying according to matrix sizes (*Saaty, 1980*).**

| n | 1 | 2 | 3 | 4 | 5 | 6 | 7 | 8 | 9 | 10 | 11 | 12 | 13 | 14 | 15 |
|---|---|---|---|---|---|---|---|---|---|----|----|----|----|----|----|
| RI | 0.00 | 0.00 | 0.58 | 0.90 | 1.12 | 1.24 | 1.32 | 1.41 | 1.45 | 1.49 | 1.51 | 1.48 | 1.56 | 1.57 | 1.59 |

(*Akıncı, Özalp & Turgut, 2012*). A normalized comparison matrix is obtained by dividing the column values of the matrix by the sum of each column. The row values obtained from the binary comparisons are added together, and the weight values are obtained by dividing the total value by the value in the row. While making pairwise comparisons of parameters in the AHP method, a certain amount of inconsistency may occur because it is made with individual or simple decisions. Therefore, the obtained priority vector should be checked for consistency (*Mezughi et al., 2012*; *Öztürk & Batuk, 2010*). The consistency index (CI) is calculated according to Equation (3).

$$CI = \frac{\lambda max - n}{n - 1} \tag{3}$$

The CR (Consistency Ratio) formula is used to calculate the consistency ratio of the binary comparison provisions (*Saaty, 1994*). In order to obtain the Consistency Ratio (CR), it is necessary to know the Random Consistency Index (RI). This index is given as the mean of the constancy index as a result, according to the order of the matrix presented by *Saaty (1980)* (Table 6).

The consistency ratio (CR) is calculated by dividing the obtained CI by the RI value (Equation (4)):

$$CR = \frac{CI}{RI} \tag{4}$$

The calculated consistency ratio must be 0.10 (10%) or less to be valid. However, if the consistency ratio is greater than 0.10, the pairwise comparison matrices should be revised (Saaty, 1980). These calculations increase the accuracy of the decisions and ensure the reliability of the analysis results (Arslan & Khisty, 2005; Saaty, 2008).

**The relationship between soil quality index classes and yield values**

The accuracy of the land suitability model, which reveals the suitability of the lands for wheat production, was carried out using the wheat yield of 2020 obtained from the Farmer Registration System and the results of the face-to-face surveys with the breeders. In determining the accuracy of the land suitability map, the relationship between the 64 yield values and the land suitability index score obtained by AHP and GIS methods is determined by regression. The ($R^2$) obtained using Equation (5) provides information about the success of the model. $R^2$ value of 1 or close to 1 gives information about the accuracy of the regression (Zhang et al., 2015).

$$R^2 = 1 - \frac{\sum_{1=1}^{n}(y_i - \overline{y})^2}{\sum_{i=1}^{n}(y_i - \hat{y}_i)^2} \tag{5}$$

Here, yi and $\hat{y}_i$ denote the observed and predicted soil values for wheat-barley yield, respectively. $\overline{y}$ represents the average of the wheat-barley yield observation values.

# RESULTS AND DISCUSSION

## Soil properties and maps

Descriptive statistical properties of the 260 surface soil samples taken from the study area are given in Table 7. The clay content of the surface soils in the study area varied between 10.53 and 60.83%, with an average of 28.62%. Considering the silt content, the lowest value was 10.54%, while the highest value was 44.95%. The sand content of the soils in the study area varies between 10.54 and 76.80%, and the average sand content was 46.73%. The hydraulic conductivity values were 1.57 cm h$^{-1}$ on average, the average bulk density value was 1.49 g cm$^{-3}$, and the available water content was 11.10% on average. The slope values varied between 1 and 30%. The soil depth was found to be between 20 and 120 cm. OM content was determined as 0.54% in wheat-barley cultivation areas and 4.04% in pastures. The lime content of the soils varied between 1.23% and 43.67%, and the average lime content was 11.24%. The pH value ranged from neutral (7.09) to strongly alkaline (9.82), and the average pH value (8.27) was determined to be slightly alkaline. The total nitrogen content varied between 0.036 and 0.540%, with an average value of 0.140%. The available phosphorus content ranged from 0.017 to 24.16 mg kg$^{-1}$, with an average phosphorus content of 5.62 mg kg$^{-1}$. The exchangeable calcium content of the soils varied between 7.17 and 40.07 cmol kg$^{-1}$, exchangeable magnesium content between 0.16 and 10.39 cmol kg$^{-1}$, exchangeable potassium content between 0.29 and 4.42 cmol kg$^{-1}$, and exchangeable
**Table 7  Descriptive statistics of the physical, chemical, and productivity properties of the soils.**

| Parameter | Min | Max | Mean | SD | CV | Skewness | Kurtosis |
|---|---|---|---|---|---|---|---|
| Clay (%) | 10.53 | 60.83 | 28.62 | 7.44 | 26.00 | 0.41 | 1.17 |
| Silt (%) | 10.54 | 44.95 | 24.65 | 6.42 | 26.04 | 0.21 | 0.91 |
| Sand (%) | 10.54 | 76.8 | 46.73 | 11.98 | 25.64 | −0.87 | −0.56 |
| HC (cm h$^{-1}$) | 0.02 | 18.75 | 1.57 | 2.57 | 0.16 | 3.58 | 16.73 |
| BD (g cm$^{-3}$) | 0.95 | 1.87 | 1.49 | 0.16 | 10.74 | −0.69 | 0.67 |
| AWC (%) | 2.69 | 18.59 | 11.10 | 2.6 | 23.42 | −0.53 | 0.41 |
| Slope (%) | 1 | 30 | 5.10 | 6.89 | 135.10 | 2.17 | 3.58 |
| Depth (cm) | 20 | 120 | 74.46 | 38.53 | 51.75 | −0.29 | −1.42 |
| pH | 7.09 | 9.82 | 8.27 | 0.38 | 4.59 | −0.50 | 1.66 |
| EC (dS m$^{-1}$) | 0.042 | 1.267 | 0.19 | 0.19 | 100.00 | 3.84 | 15.84 |
| OM (%) | 0.54 | 4.04 | 1.65 | 0.82 | 49.70 | 0.96 | 0.29 |
| CaCO3 (%) | 1.23 | 43.67 | 11.24 | 9.08 | 80.78 | 1.26 | 1.60 |
| CEC (cmol kg$^{-1}$) | 9.63 | 47.43 | 30.03 | 8.90 | 29.64 | −0.31 | −0.55 |
| ESP (%) | 0.80 | 11.49 | 4.29 | 1.83 | 42.66 | 1.18 | 1.57 |
| N (%) | 0.036 | 0.540 | 0.140 | 0.082 | 58.57 | 1.56 | 2.95 |
| P (mg kg$^{-1}$) | 0.017 | 24.16 | 5.62 | 2.47 | 43.95 | 3.95 | 23.77 |
| Ca (cmol kg$^{-1}$) | 7.17 | 40.07 | 25.14 | 7.63 | 30.35 | −0.39 | −0.50 |
| Mg (cmol kg$^{-1}$) | 0.16 | 10.39 | 2.47 | 2.03 | 82.19 | 1.63 | 2.57 |
| K (cmol kg$^{-1}$) | 0.29 | 4.42 | 1.19 | 0.53 | 44.54 | 1.70 | 5.99 |
| Na (cmol kg$^{-1}$) | 0.15 | 5.31 | 1.24 | 0.62 | 50.00 | 2.72 | 12.33 |
| Fe (mg kg$^{-1}$) | 2.57 | 77.83 | 11.85 | 9.41 | 79.41 | 3.42 | 14.81 |
| Zn (mg kg$^{-1}$) | 0.16 | 5.21 | 0.61 | 0.63 | 103.01 | 3.77 | 17.93 |
| Cu (mg kg$^{-1}$) | 0.04 | 5.65 | 1.34 | 0.77 | 57.46 | 2.18 | 7.81 |
| Mn (mg kg$^{-1}$) | 4.78 | 172.54 | 33.95 | 21.91 | 64.54 | 2.77 | 11.40 |

sodium content between 0.15 and 5.31 cmol kg$^{-1}$. The Fe, Zn, Cu, and Mn contents of the soils vary between 2.57 and 77.83 mg kg$^{-1}$, 0.16 and 5.21 mg kg$^{-1}$, 0.04 and 5.65 mg kg$^{-1}$, and 4.78 and 172.54 mg kg$^{-1}$, respectively. The average available Fe, Zn, Cu, and Mn contents were determined as 11.85, 0.61, 1.34, and 33.95 mg kg$^{-1}$, respectively.

The purpose of giving descriptive statistics is to emphasize the determined value ranges of the criteria examined, to express the changes within the variation with the coefficient of variation, and to make the criteria examined understandable. The investigated soil properties are important in defining and classifying the soil. It constitutes a selection criterion in the selection of cultivable plant patterns on the identified and classified soil structures and in the suitability of their cultivation. For this reason, the criteria we examine in land suitability studies are accepted as basic research criteria (*Bayraklı, Dengiz & Kars, 2023*). In determining the suitability of wheat-barley agriculture, the limit values required by the plant regarding the soil are known. Considering the score values of the examined criteria, land suitability for wheat-barley cultivation is determined.

The coefficients of variation of the surface soils of the study area are given in Table 7. Hydraulic conductivity, bulk density, and pH had coefficients of variation less than 15%. Sand, clay, silt, available water capacity, CEC, and exchangeable Ca were observed to be

**Table 8  Root of square error (RMSE) values of the interpolation methods of the physical properties of the surface soils of the study area.**

| | Distribution models | | Clay | Silt | Sand | HC | BD (g cm$^{-3}$) | AWC | Slope | Depth (cm) |
|---|---|---|---|---|---|---|---|---|---|---|
| | | | (%) | | | | | (%) | | |
| **IDW** | 1 | | 6.4165 | 6.0081 | 10.613 | 2.455 | 0.1556 | 2.348 | 3.058 | 19.1083 |
| | 2 | | 6.3112 | 5.9833 | 10.47 | 2.4665 | 0.1559 | 2.3464 | 2.9036 | 18.0792 |
| | 3 | | 6.2709 | 5.9912 | 10.414 | 2.4879 | 0.1699 | 2.3601 | 2.7994 | 17.32 |
| **RBF** | TPS | | 6.7882 | 6.81 | 11.497 | 2.84 | 0.183 | 2.8321 | 2.6306 | 16.2787 |
| | CRS | | 6.2473 | 6.0562 | 10.471 | 2.5518 | 0.159 | 2.3901 | 2.5623 | 16.0184 |
| | ST | | 6.2461 | 6.011 | 10.445 | 2.5144 | 0.157 | 2.368 | 2.5977 | 16.1243 |
| **Kriging** | Ordinary | Gaussian | 6.2986 | 6.0151 | 10.5462 | 2.6593 | 0.158 | 2.3546 | 2.9262 | 19.3853 |
| | | Spherical | 6.2539 | 5.9842 | 10.535 | 2.6315 | 0.1571 | 2.3821 | 2.8361 | 17.0732 |
| | | Exponantial | 6.2752 | 6.028 | 10.4045 | 2.6312 | 0.1571 | 2.3521 | 3.0027 | 16.437 |
| | Simple | Gaussian | 6.2613 | 6.0041 | 10.5146 | 2.5909 | 0.1574 | 2.347 | 2.9116 | 21.2923 |
| | | Spherical | 6.2352 | 6.0161 | 10.451 | 2.6061 | 0.1569 | 2.3464 | 2.7898 | 18.1135 |
| | | Exponantial | 6.2376 | 5.979 | 10.472 | 2.57 | 0.157 | 2.347 | 2.7735 | 16.4471 |
| | Universal | Gaussian | 6.2985 | 6.015 | 10.5461 | 2.6589 | 0.1581 | 2.3603 | 2.9262 | 19.3853 |
| | | Spherical | 6.253 | 5.9801 | 10.4049 | 2.6308 | 0.1578 | 2.3801 | 3.0027 | 17.0732 |
| | | Exponantial | 6.2752 | 6.0281 | 10.5356 | 2.63 | 0.1578 | 2.3521 | 2.8361 | 16.4369 |

between 15 and 35%. It has been observed that the slope, depth, EC, OM, lime, ESP, N, P, Mg, K, Na, Fe, Zn, Cu, and Mn values of the soils have coefficients of variation over 35%. Sand, bulk density, and available water showed negative skewness to the left, while hydraulic conductivity and slope showed positive skewness to the right. Normal distribution was observed in pH and CEC. Chemical parameters, EC, OM, CaCO$_3$, and ESP, showed positive skewness to the right. The soil fertility properties N, P, Mg, K, Na, Fe, Zn, Cu, and Mn showed positive skewness to the right. Exchangeable Ca content showed a normal distribution. Statistically, 70.83% of all soil parameters examined in the study area show a positive distribution skewed to the right. In a left-skewed, negative distribution, the features have a higher distribution frequency than the mean, while the opposite happens in a right-skewed, positive distribution (*Pacci, Saflı& Dengiz, 2023*).

Different interpolation methods were examined in the study. The performances of the models, which have different approaches to spatial distribution, were evaluated. Spatial distribution maps were produced by selecting the highest-performing models. The distribution of SQI was examined using different interpolation methods, and the RMSE values are given in Tables 8, 9 and 10. RMSE values are an important criterion in the selection of models. The smaller the RMSE value, the more accurate the model is (*Şenol et al., 2020*; *Alaboz, Odabas & Dengiz, 2023*). Fifteen models were used in our study. For each soil criterion, the model representing the smallest RMSE value was selected, and a map was made for that parameter. For each soil criterion examined, six maps were produced using the completely regularized spline (CRS) method whose RMSE values were at most radial based function (RBF).

**Table 9  Root mean square error (RMSE) values of the interpolation methods of the chemical properties of the surface soils of the study area.**

| | Distribution models | | pH | EC (dS m$^{-1}$) | O.M (%) | CaCO$_3$ (%) | KDK (cmol kg$^{-1}$) | ESP (%) |
|---|---|---|---|---|---|---|---|---|
| **IDW** | | 1 | 0.3189 | 0.1817 | 0.6743 | 7.3168 | 6.2484 | 1.771 |
| | | 2 | 0.3102 | 0.1807 | 0.6636 | 7.1724 | 6.2959 | 1.7932 |
| | | 3 | 0.3045 | 0.1804 | 0.6591 | 7.0824 | 6.3857 | 1.8275 |
| **RBF** | | TPS | 0.3235 | 0.1994 | 0.7256 | 7.478 | 7.7215 | 2.2622 |
| | | CRS | 0.3025 | 0.1812 | 0.6562 | 7.045 | 6.4812 | 1.8612 |
| | | ST | 0.3027 | 0.1805 | 0.6561 | 7.051 | 6.3912 | 1.8248 |
| **Kriging** | Ordinary | Gaussian | 0.3062 | 0.1824 | 0.6934 | 7.8365 | 6.3952 | 1.768 |
| | | Spherical | 0.3024 | 0.1816 | 0.6661 | 7.9185 | 6.7548 | 1.7827 |
| | | Exponantial | 0.3063 | 0.1821 | 0.6855 | 7.9678 | 6.4512 | 1.7715 |
| | Simple | Gaussian | 0.3267 | 0.1839 | 0.6861 | 7.9143 | 6.3396 | 1.7582 |
| | | Spherical | 0.3144 | 0.1829 | 0.6709 | 7.5835 | 6.5678 | 1.7573 |
| | | Exponantial | 0.3223 | 0.1838 | 0.69 | 7.6913 | 6.3848 | 1.7635 |
| | Universal | Gaussian | 0.3062 | 0.1826 | 0.6934 | 7.8321 | 6.3958 | 1.768 |
| | | Spherical | 0.3025 | 0.1819 | 0.6607 | 7.9818 | 6.7514 | 1.7827 |
| | | Exponantial | 0.3063 | 0.1823 | 0.6855 | 7.9647 | 6.4578 | 1.7715 |

The RMSE values of each model were determined for the physical, chemical, and productivity parameters of the study area surface soils. The RMSE values of the models are given in Tables 8, 9 and 10. Spatial distribution maps of the parameters used in determining the land suitability of the study area for wheat-barley cultivation are given in Fig. 3.

## Impact of criteria on land suitability

The study area consists of different topographic structures and geological units. Therefore, 24 different soil parameters were used to evaluate the suitability of the land. The AHP method was used to determine the relative weights of the parameters used for land suitability assessment in wheat-barley cultivation. The matrix of the pairwise comparisons of physical, chemical, productivity, and main parameters with the AHP method is given in Table 11. The normalized matrix is presented in Table 12. The CR values indicating the consistency of the AHP were calculated as 0.096, 0.090, 0.090, and 0.055, respectively (Table 13). According to the consistency results obtained, it was seen that the AHP matrices were consistent and within acceptable limits.

The weight parameters obtained as a result of the evaluation of soil properties with AHP are given in Table 14. Efficiency parameters (Hierarchy B3) were represented by the lowest core weight value of 0.1958, while the highest core weight value was 0.4934 for physical properties (Hierarchy B1). For each hierarchy, it was seen that OM, slope, and N contributed the most from physical, chemical, and productivity parameters, respectively. In terms of soil fertility and quality, the chemical properties of the soil and the amount of nutrients are at optimum levels. The physical content of the soil is not in suitable conditions. Although the chemical and fertility parameters of the soils are at the desired level, the unsuitability of the physical parameters negatively affects the soil quality index. Therefore, the contribution rate of physical quality parameters has the greatest effect

**Table 10 Root of square error (RMSE) values of the interpolation methods of the productivity properties of the study area surface soils.**

| | Distribution models | | N (%) | P (mg kg⁻¹) | K | Ca | Mg | Na | Fe | Cu | Mn | Zn |
|---|---|---|---|---|---|---|---|---|---|---|---|---|
| | | | | | | (cmol kg⁻¹) | | | (mg kg⁻¹) | | | |
| **IDW** | 1 | | 0.07 | 2.2371 | 0.4846 | 5.5176 | 1.7814 | 0.5778 | 8.7352 | 0.7627 | 19.8262 | 0.5992 |
| | 2 | | 0.0693 | 2.242 | 0.4814 | 5.5568 | 1.7461 | 0.5818 | 8.6086 | 0.753 | 19.6305 | 0.6059 |
| | 3 | | 0.0686 | 2.2545 | 0.4794 | 5.6293 | 1.7247 | 0.5898 | 8.5503 | 0.7457 | 19.5899 | 0.6161 |
| **RBF** | TPS | | 0.074 | 2.569 | 0.53 | 5.7388 | 1.8444 | 0.6986 | 9.3652 | 0.7607 | 22.222 | 0.7334 |
| | CRS | | 0.0681 | 2.2851 | 0.4791 | 6.8029 | 1.7057 | 0.5952 | 8.5768 | 0.7435 | 19.8458 | 0.6276 |
| | ST | | 0.0682 | 2.2623 | 0.4806 | 5.6408 | 1.707 | 0.587 | 8.5762 | 0.7454 | 19.7352 | 0.6162 |
| **Kriging** | Ordinary | Gaussian | 0.0717 | 2.2469 | 0.4947 | 5.5799 | 1.7923 | 0.575 | 8.6227 | 0.7534 | 19.9453 | 0.5959 |
| | | Spherical | 0.0709 | 2.2423 | 0.4895 | 5.9647 | 1.7531 | 0.5769 | 8.593 | 0.7518 | 19.9792 | 0.5962 |
| | | Exponantial | 0.0692 | 2.2433 | 0.4925 | 5.6335 | 1.7956 | 0.5746 | 8.5906 | 0.7506 | 19.8796 | 0.5968 |
| | Simple | Gaussian | 0.071 | 2.2493 | 0.4901 | 5.5411 | 1.7598 | 0.5733 | 8.9989 | 0.7509 | 20.1862 | 0.6044 |
| | | Spherical | 0.0707 | 2.2669 | 0.4837 | 5.7277 | 1.7457 | 0.5715 | 8.9705 | 0.7422 | 20.1423 | 0.6045 |
| | | Exponantial | 0.0689 | 2.2475 | 0.4889 | 5.5833 | 1.7692 | 0.5726 | 8.9428 | 0.7516 | 20.0464 | 0.605 |
| | Universal | Gaussian | 0.0717 | 2.2469 | 0.4947 | 5.5799 | 1.7923 | 0.575 | 8.6227 | 0.7534 | 19.9453 | 0.5985 |
| | | Spherical | 0.0709 | 2.2424 | 0.4895 | 5.9647 | 1.7531 | 0.5769 | 8.593 | 0.7518 | 19.9792 | 0.5962 |
| | | Exponantial | 0.0692 | 2.2433 | 0.4925 | 5.6335 | 1.7956 | 0.5746 | 8.5966 | 0.7506 | 19.8796 | 0.5968 |

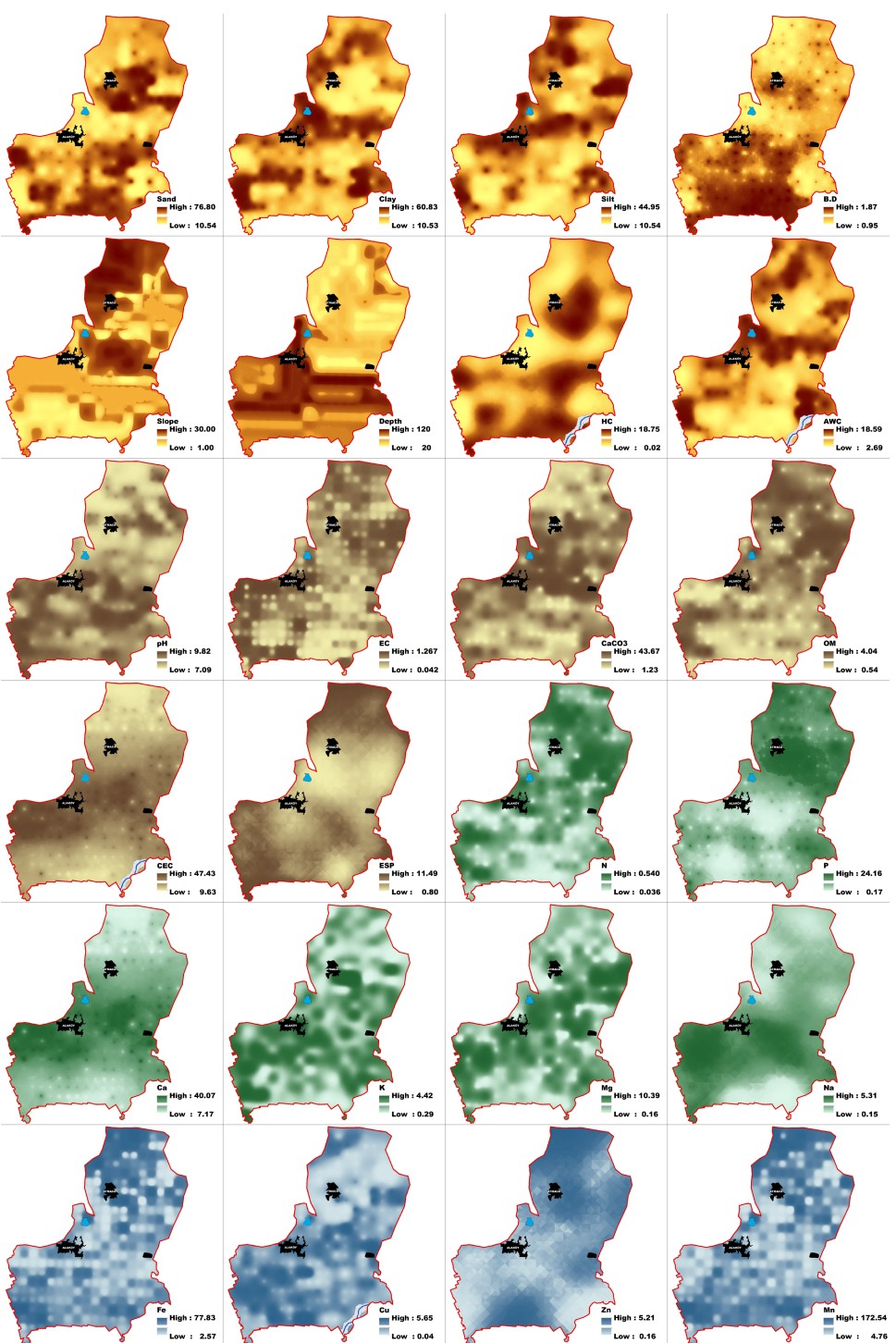

**Figure 3   Spatial distribution maps of physical, chemical and productivity parameters.**

**Table 11   Pairwise comparison matrix of physical, chemical, productivity and main parameters.**

| Physical parameters | Clay | Silt | Sand | HC | BD | UW | Slope | Depth | | |
|---|---|---|---|---|---|---|---|---|---|---|
| Clay | 1 | 3 | 3 | 2 | 2 | 1/2 | 1/3 | 1/2 | | |
| Silt | 1/3 | 1 | 1/3 | 1/5 | 1/5 | 1/5 | 1/3 | 1/3 | | |
| Sand | 1/3 | 3 | 1 | 1/3 | 1/3 | 1/3 | 1/3 | 1/3 | | |
| HC | 1/2 | 5 | 3 | 1 | 1/3 | 1/5 | 1/2 | 1/3 | | |
| BD | 1/2 | 5 | 3 | 3 | 1 | 1/3 | 1/3 | 1/2 | | |
| AWC | 2 | 5 | 3 | 5 | 3 | 1 | 1/2 | 1/2 | | |
| Slope | 3 | 3 | 3 | 2 | 3 | 2 | 1 | 3 | | |
| Depht | 2 | 3 | 3 | 3 | 2 | 2 | 1/3 | 1 | | |
| **Chemical parameters** | **pH** | **EC** | **OM** | **CaCO3** | **CEC** | | **ESP** | | | |
| pH | 1 | 3 | 1/2 | 3 | 1/3 | | 3 | | | |
| EC | 1/3 | 1 | 1/3 | 1/3 | 1/3 | | 1/3 | | | |
| OM | 2 | 3 | 1 | 3 | 2 | | 3 | | | |
| CaCO3 | 1/3 | 3 | 1/3 | 1 | 1/3 | | 1/3 | | | |
| CEC | 3 | 3 | 1/2 | 3 | 1 | | 3 | | | |
| ESP | 1/3 | 3 | 1/3 | 3 | 1/3 | | 1 | | | |
| **Productivity parameters** | **N** | **P** | **K** | **Ca** | **Mg** | **Na** | **Fe** | **Cu** | **Zn** | **Mn** |
| N | 1 | 2 | 3 | 3 | 5 | 7 | 5 | 5 | 3 | 5 |
| P | 1/2 | 1 | 3 | 5 | 7 | 7 | 3 | 3 | 3 | 5 |
| K | 1/3 | 1/3 | 1 | 3 | 3 | 7 | 3 | 3 | 3 | 5 |
| Ca | 1/3 | 1/5 | 1/3 | 1 | 2 | 3 | 1/5 | 1/3 | 1/5 | 1/3 |
| Mg | 1/5 | 1/7 | 1/3 | 1/2 | 1 | 3 | 1/5 | 1/3 | 1/5 | 1/3 |
| Na | 1/7 | 1/7 | 1/7 | 1/3 | 1/3 | 1 | 1/5 | 1/3 | 1/3 | 1/3 |
| Fe | 1/5 | 1/3 | 1/3 | 5 | 5 | 5 | 1 | 3 | 3 | 3 |
| Cu | 1/5 | 1/3 | 1/3 | 3 | 3 | 3 | 1/3 | 1 | 1/3 | 3 |
| Zn | 1/3 | 1/3 | 1/3 | 5 | 5 | 3 | 1/3 | 3 | 1 | 3 |
| Mn | 1/5 | 1/5 | 1/5 | 3 | 3 | 3 | 1/3 | 1/3 | 1/3 | 1 |
| **Main criteria** | **Physically** | | **Chemical** | | **Productivity** | | | | | |
| Physically | 1 | | 2 | | 2 | | | | | |
| Chemical | 0.5 | | 1 | | 2 | | | | | |
| Productivity | 0.5 | | 0.5 | | 1 | | | | | |
| Total | 2 | | 3.5 | | 5 | | | | | |

(*Şenol et al., 2020*). Among the determined physical parameters, the most weight value was sloped. The slope factor affects most agricultural activities, irrigation methods, erosion rate, agricultural input use, and soil quality (*Mihoub et al., 2022*). Therefore, as the percentage slope decreases, it positively affects the productivity of products such as wheat-barley (*Seyedmohammadi et al., 2019*). While soil organic matter was represented with the highest weight (0.2982) among the parameters, exchangeable sodium (Na) had the lowest weight with a value of 0.0192 (Table 14). Therefore, in terms of productivity, OM is accepted as one of the most important quality parameters in obtaining soil quality index (*Andrews, Karlen & Cambardella, 2004*). Organic matter is an important indicator of soil fertility and is of great importance for agricultural production and food security

**Table 12  Normalized comparison matrix of physical, chemical, productivity and main parameters.**

| Physical parameters | Clay | Silt | Sand | HC | BD | UW | Slope | Depth | | |
|---|---|---|---|---|---|---|---|---|---|---|
| Clay | 0.10 | 0.11 | 0.16 | 0.12 | 0.17 | 0.08 | 0.09 | 0.08 | | |
| Silt | 0.03 | 0.04 | 0.02 | 0.01 | 0.02 | 0.03 | 0.09 | 0.05 | | |
| Sand | 0.03 | 0.11 | 0.05 | 0.02 | 0.03 | 0.05 | 0.09 | 0.05 | | |
| HC | 0.05 | 0.18 | 0.16 | 0.06 | 0.03 | 0.03 | 0.14 | 0.05 | | |
| BD | 0.05 | 0.18 | 0.16 | 0.18 | 0.08 | 0.05 | 0.09 | 0.08 | | |
| AWC | 0.21 | 0.18 | 0.16 | 0.30 | 0.25 | 0.15 | 0.14 | 0.08 | | |
| Slope | 0.10 | 0.11 | 0.16 | 0.12 | 0.17 | 0.08 | 0.09 | 0.08 | | |
| Depht | 0.03 | 0.04 | 0.02 | 0.01 | 0.02 | 0.03 | 0.09 | 0.05 | | |
| **Chemical parameters** | **pH** | **EC** | | **OM** | **CaCO3** | **CEC** | | **ESP** | | |
| pH | 0.14 | 0.19 | | 0.17 | 0.23 | 0.08 | | 0.28 | | |
| EC | 0.05 | 0.06 | | 0.11 | 0.03 | 0.08 | | 0.03 | | |
| OM | 0.29 | 0.19 | | 0.33 | 0.23 | 0.46 | | 0.28 | | |
| CaCO3 | 0.05 | 0.19 | | 0.11 | 0.08 | 0.08 | | 0.03 | | |
| CEC | 0.43 | 0.19 | | 0.17 | 0.23 | 0.23 | | 0.28 | | |
| ESP | 0.05 | 0.19 | | 0.11 | 0.23 | 0.08 | | 0.09 | | |
| **Productivity  parameters** | **N** | **P** | **K** | **Ca** | **Mg** | **Na** | **Fe** | **Cu** | **Zn** | **Mn** |
| N | 0.34 | 0.45 | 0.35 | 0.14 | 0.19 | 0.19 | 0.39 | 0.31 | 0.21 | 0.19 |
| P | 0.17 | 0.22 | 0.35 | 0.24 | 0.27 | 0.19 | 0.23 | 0.19 | 0.21 | 0.19 |
| K | 0.12 | 0.07 | 0.12 | 0.14 | 0.11 | 0.19 | 0.23 | 0.19 | 0.21 | 0.19 |
| Ca | 0.12 | 0.05 | 0.04 | 0.05 | 0.08 | 0.08 | 0.02 | 0.02 | 0.01 | 0.01 |
| Mg | 0.07 | 0.03 | 0.04 | 0.02 | 0.04 | 0.08 | 0.02 | 0.02 | 0.01 | 0.01 |
| Na | 0.05 | 0.03 | 0.02 | 0.02 | 0.01 | 0.03 | 0.02 | 0.02 | 0.02 | 0.01 |
| Fe | 0.07 | 0.07 | 0.04 | 0.24 | 0.19 | 0.14 | 0.08 | 0.19 | 0.21 | 0.12 |
| Cu | 0.07 | 0.07 | 0.04 | 0.14 | 0.11 | 0.08 | 0.03 | 0.06 | 0.02 | 0.12 |
| Zn | 0.12 | 0.07 | 0.04 | 0.24 | 0.19 | 0.08 | 0.03 | 0.19 | 0.07 | 0.12 |
| Mn | 0.07 | 0.05 | 0.02 | 0.14 | 0.11 | 0.08 | 0.03 | 0.02 | 0.02 | 0.04 |
| **Main criteria** | **Physically** | | | **Chemical** | | | **Productivity** | | | |
| Physically | 0.50 | | | 0.57 | | | 0.40 | | | |
| Chemical | 0.25 | | | 0.29 | | | 0.40 | | | |
| Productivity | 0.25 | | | 0.14 | | | 0.20 | | | |
| Total | 1.00 | | | 1.00 | | | 1.00 | | | |

**Table 13  Consistency rate of AHP.**

| Main criteria | Physically | Chemical | Productivity | Total |
|---|---|---|---|---|
| Max. Eigenvalue | 8.942625 | 6.569387 | 6.569387 | 3.053618 |
| Consistency Index (CI) | 0.135513 | 0.11264 | 0.135350 | 0.032449 |
| Random Index (RI) | 1.41 | 1.24 | 1.49 | 0.58 |
| Consistency Ratio (CR) | 0.096109 | 0.090839 | 0.090839 | 0.055946 |

**Table 14  AHP weighting of soil quality indicators of the study area.**

| Hierarchy A | Hierarchy B | | | |
| --- | --- | --- | --- | --- |
| | Physically B1 | Chemical B2 | Productivity B3 | |
| Hierarchy C | 0.4934 | 0.3108 | 0.1958 | Weight |
| Clay | 0.1131 | | | 0.0558 |
| Silt | 0.0342 | | | 0.0169 |
| Sand | 0.0513 | | | 0.0253 |
| HC | 0.0804 | | | 0.0397 |
| BD | 0.1078 | | | 0.0532 |
| AWC | 0.1875 | | | 0.0925 |
| Slope | 0.2497 | | | 0.1232 |
| Depth | 0.176 | | | 0.0868 |
| pH | | 0.1812 | | 0.0563 |
| EC | | 0.0565 | | 0.0176 |
| OM | | 0.2982 | | 0.0927 |
| CaCO3 | | 0.082 | | 0.0255 |
| CEC | | 0.2631 | | 0.0818 |
| ESP | | 0.119 | | 0.0370 |
| N | | | 0.2548 | 0.0499 |
| P | | | 0.2052 | 0.0402 |
| K | | | 0.1468 | 0.0287 |
| Ca | | | 0.0356 | 0.0070 |
| Mg | | | 0.0274 | 0.0054 |
| Na | | | 0.0192 | 0.0038 |
| Fe | | | 0.114 | 0.0223 |
| Cu | | | 0.062 | 0.0121 |
| Zn | | | 0.091 | 0.0178 |
| Mn | | | 0.044 | 0.0086 |
| Total | 1 | 1 | 1 | 1.0000 |

(*Obalum et al., 2017*; *Mihoub et al., 2022*). The positive effects of organic matter on the physical and chemical properties of the soil have been reported in many studies (*Erdal et al., 2018*; *Alaboz et al., 2021*).

The total percent nitrogen content of soils has a very important role in determining soil fertility scores (*Dadhich, Patel & Kalubarme, 2017*; *Günal et al., 2022*). The high level of exchangeable Ca and Na contents causes the productivity parameters to have low weight values in the created hierarchy. It has been observed both in our study and in many literatures that the high pH of the soil causes problems in the chemical quality scores of the soil. For the healthy growth of wheat-barley, it is desirable that the soil pH level be neutral or close to neutral. That is, soil pH is important for the healthy development of wheat-barley (*Mandal et al., 2020*; *Pilevar et al., 2020*; *Mihoub et al., 2022*). It is reported that as the exchangeable sodium content of the soil increases, it has negative effects on the physical structure and permeability level of the soil (*Boyer et al., 2008*; *Pilevar et al.,*

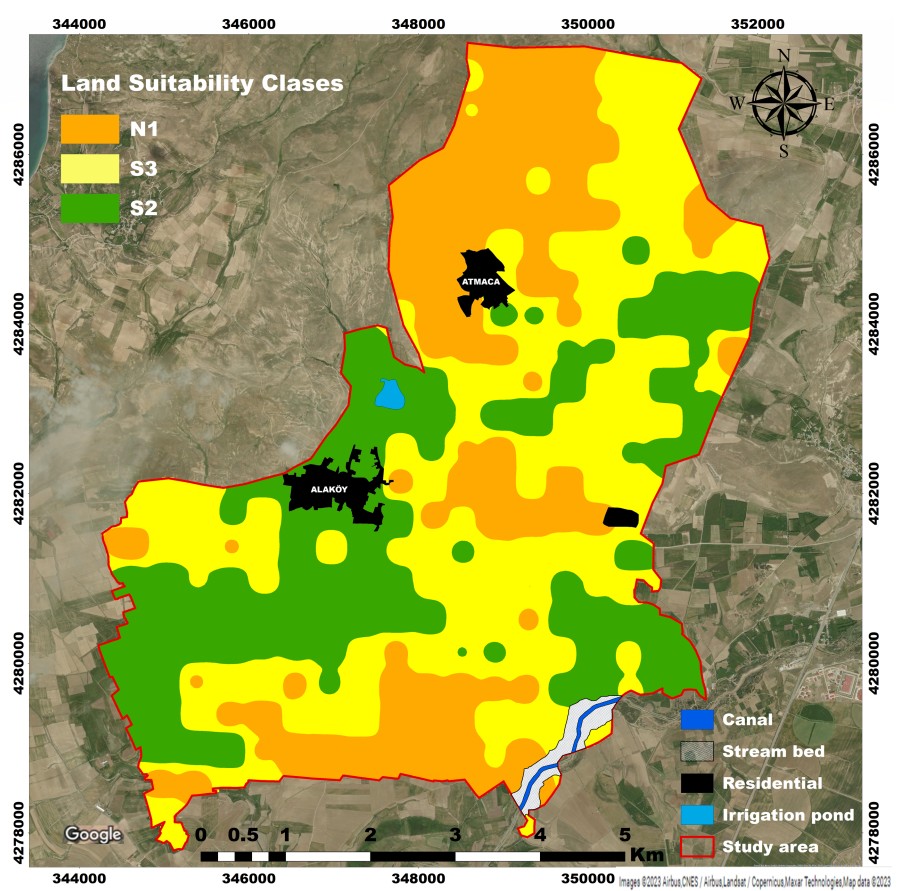

**Figure 4  Land suitability map for wheat-barley production.** Satellite image credit: https://www.google.com/maps.

*2020*; *Sparks, Singh & Siebecker, 2023*). Fe affects yield and quality positively (*Havlin, 2020*). While Cu in the soil is found to be sufficient for plants at low contents, Cu in excessive amounts in the soil has a phytotoxic effect on plants (*Michaud et al., 2007*). It has been found that sufficient Zn increases the yield and quality of wheat-barley (*Hui et al., 2022*). Chlorosis (yellowing) was observed between leaf veins in plants with Mn deficiency (*McCauley, Jones & Jacobsen, 2009*; *Kacar & Katkat, 2010*).

## Land suitability assessment for wheat-barley production

The suitability of the study area for wheat-barley cultivation is presented in Fig. 4. The resulting map was created with the AHP-GIS hybrid approach. Areas suitable for wheat-barley cultivation are presented in Table 15. A percentage of 28.29% (1189.94 ha) of the studied area was found to be well suited (S2) for wheat-barley production. Moderately suitable (S3) areas cover 1679.95 ha with 39.93%. Weak (N1) suitable areas were found to be 1185.20 ha with a rate of 28.17%. 2.29% (96.49 ha) of the study area was determined as a settlement, 0.94% (39.62 ha) as a stream bed, 0.21% (8.79 ha) as an irrigation pond and 0.16% (6.83 ha) as an irrigation canal.

**Table 15  Spatial distribution of classes in the suitability map.**

| Suitability | Area (ha) | Ratio (%) |
| --- | --- | --- |
| N1 | 1185.20 | 28.17 |
| S3 | 1679.95 | 39.93 |
| S2 | 1189.94 | 28.29 |
| Residential | 96.49 | 2.29 |
| Canal | 6.83 | 0.16 |
| Stream bed | 39.62 | 0.94 |
| İrrigation pond | 8.79 | 0.21 |
| Total | 4206.82 | 100 |

When the distribution map representing the soil quality index values was examined, it was seen that the regions with the lowest quality (N1) in the study area were located in the northwest and south directions (Table 15). The main reason for this situation is due to the high level of slope, sand, and silt content of the soil in the northwest and south directions. Moderately suitable areas (S3) were spread over the western and eastern edges of the study area. This is due to the fact that the physical parameters of the soils representing the well-suited (S2) category in the study area are represented with the highest weight values. Soil texture significantly affects soil aeration, water holding capacity, nutrient content, soil aggregation, structure development, and tillage (*Dharumarajan & Hegde, 2022*). The value of 68.22% (S2+S3) of the surveyed land represents the acceptable group in wheat-barley farming. When the land structure was examined, the soils clayey loam, silty-clay loam, sandy-clay-loam, silty, silty-loam, silty-clay, and sandy-clay texture classes were also observed (Table 15). This structure is suitable for optimum development of wheat-barley plants. Soil texture is important in wheat-barley cultivation. For example, loamy soils protect the plant against drought because of their high water holding capacity, while sandy soils provide good aeration and regulate the temperature of the soil. Clay soils are important in terms of soil structure and aggregation. It is also effective in the retention and utilization of plant nutrients (*Pilevar et al., 2020*; *Mandal et al., 2020*; *Mihoub et al., 2022*).

The increase in bulk density due to the increase in the amount of sand in the soil is a negative situation regarding root development (*Pacci et al., 2022*). The increase in the clay content of the soil and the amount of OM has a positive effect on the water retention in the soil. Furthermore, it has a positive effect on the physical quality index. Both are very important in water conservation. The increase in the available water content of the soil is a significant variable in plant development. The increased amount of available water proves that the plant can benefit from more water (*Alaboz, Demir & Dengiz, 2020*).

The land suitability assessment has been identified as one of the scientific planning and managerial approaches to promote the sustainability of agricultural production. It is known that many researchers use this model similarly in land suitability assessment studies (*Zhang et al., 2015*; *Ostovari et al., 2019*; *Orhan, 2021*; *Talukdar et al., 2022*).
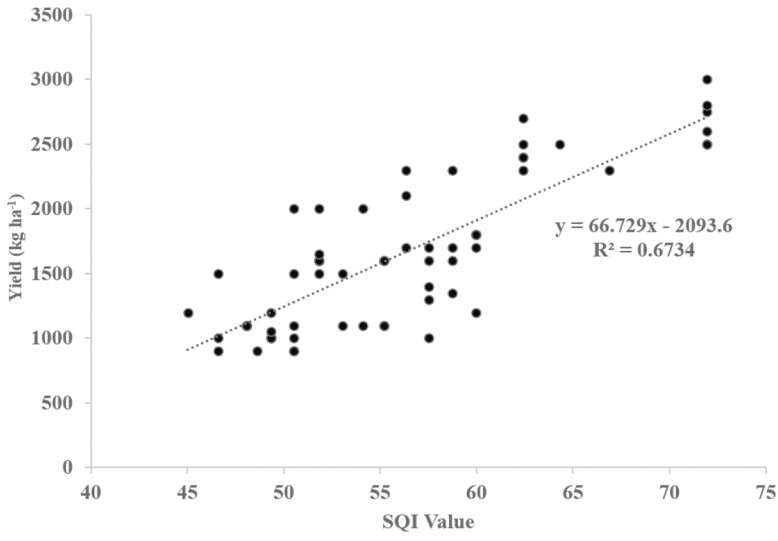

**Figure 5** Regression graph between soil quality index (SQI) values and productivity values.

## Validation of the land suitability for wheat-barley cultivation

In the study area, wheat-barley yield for 2020 varies between 900 and 3,000 kg ha$^{-1}$, and the average yield in 2020 was 1,600 kg ha$^{-1}$. The relationship between wheat-barley yield grown in Alaköy and Atmaca and SQI was assessed using regression analysis. The regression coefficient value was found to be $R^2 = 0.67$. This reveals the suitability of the model and shows that the prediction level is successful (Fig. 5). In addition, *Sys Ir et al. (1993)* reported that grain production of 2,500–3,500 kg ha$^{-1}$ in rainy conditions and 4,000–6,000 kg ha$^{-1}$ in irrigation conditions is a good commercial yield. The average yield value was taken since wheat-barley grown in the studied study area was grown under different conditions (wet and dry). It shows that the validation results and the land suitability analysis using AHP and GIS integration were in harmony. Several researchers reported that the AHP method offers more realistic and rational evaluations in the analysis of hierarchical and complex data. At the same time, it is stated that AHP has many more effective and successful advantages than other classical parametric methods (*Yalew, Van Griensven & Van der Zaag, 2016*; *Dedeoglu & Dengiz, 2019*; *Orhan, 2021*).

## CONCLUSION

This study reveals that the methods and modeling used to determine land suitability classes are highly usable in production planning. Determining land suitability classes in agricultural lands is considered an important achievement in the effective and sustainable use of land for wheat-barley cultivation. It is seen that it is critical to make land suitability classifications in land and watershed management, planning, and use. Firstly, it is necessary to determine land suitability classes in agricultural lands. Thus, by ensuring the sustainability of land use, it may be possible to protect plant species and diversity by ensuring ecological balance. For this purpose, the effects of the parameters on land suitability classes were weighted

using the AHP model, and the standard scoring function was used. The soil quality map created by this method includes three different soil quality index classes as poor, medium, and good. A positive regression ($R^2 = 0.67$) was found between the soil quality index and the yield values obtained from farmer data records and survey data.

This level of relationship shows the effectiveness of the modeling. According to the suitability analysis, it was determined that 28.29% of the study area is well (S2) suitable, 39.93% moderately (S3) suitable, and 28.17% poorly (N1) suitable for wheat-barley production. These results will benefit farmers in practice and have a positive socio-economic impact. This study will enable the preparation of agriculturally sustainable management plans and the increase of agricultural production by protecting natural resources. At the same time, it is foreseen that it can increase the efficiency of land consolidation studies, product-specific land suitability assessments, and land integration.

Making land suitability classifications is important in determining which land falls into which suitability classes during the production phase, in choosing an effective product pattern, and in economic production. With this study, it was determined which part of the region was suitable for wheat-barley production. It is inevitable that there will be economic losses as a result of insisting on wheat-barley cultivation in areas with low suitability classes.

Land use planning is possible with such studies. The study can be applied as a spatial decision support system in agricultural planning, and effective and economic agricultural planning can be provided. In addition, the obtained parameters are essential for both field and horticultural agriculture. It will be useful in planning the economic cultivation of all cropping patterns. It can be considered a basic study for different cropping patterns other than wheat-barley cultivation.

### Funding
The present study was supported by the Scientific Research Projects Coordination Unit of Van Yüzüncü Yıl University as a doctoral project numbered FDK-2020-8962. The funders had no role in study design, data collection and analysis, decision to publish, or preparation of the manuscript.

### Grant Disclosures
The following grant information was disclosed by the authors:
The Scientific Research Projects Coordination Unit of Van Yüzüncü Yıl University: FDK-2020-8962.

### Competing Interests
The authors declare there are no competing interests.

### Author Contributions
- Bulut Sargın conceived and designed the experiments, performed the experiments, analyzed the data, prepared figures and/or tables, authored or reviewed drafts of the article, and approved the final draft.

- Siyami Karaca conceived and designed the experiments, performed the experiments, analyzed the data, prepared figures and/or tables, authored or reviewed drafts of the article, and approved the final draft.

### Data Availability

The raw measurements are available in the Supplemental Files.

### Supplemental Information

Supplemental information for this article can be found online at http://dx.doi.org/10.7717/peerj.16396#supplemental-information.

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
