# Peer review of "Land suitability assessment for wheat-barley cultivation in a semi-arid region of Eastern Anatolia in Turkey"

_PeerJ, doi:10.7717/peerj.16396_

## Round 0.1 · original submission · Major Revisions

Please incorporate all the comments of the reviewers and please submit the revised version along with a point-to-point rebuttal letter. Please also improve the language of the paper.

**Language Note:** The Academic Editor has identified that the English language must be improved. PeerJ can provide language editing services - please contact us at [email protected] for pricing (be sure to provide your manuscript number and title). Alternatively, you should make your own arrangements to improve the language quality and provide details in your response letter. – PeerJ Staff

·

Basic reporting

Abstract: Very good study. Authors reported that "the study provides valuable insights into the use of advanced techniques in land suitability studies for crop cultivation" in line 31. What are those insight. Write a line on it.
Introduction: Accept my appreciation for writing good introduction. However, you needed to address following questions to improve the quality of paper. You needed to address, why you selected this topic to study? why it is important to conduct this study? What are the problems? What will happen if such studies will not be conducted? Write a complete statement of problem in the last paragraph of introduction.

Experimental design

Accept my appreciation for writing good methodology. You described data well.

Validity of the findings

Results obtained as in methods and material discussed however these results are not justified from past literature. Discuss more on it instead of explaining numbers.
Conclusion: Policy implications should be from results obtained and not from imagination. Then, put these in the last lines of abstract (That you wrote some insights).

Reviewer 2 ·

Basic reporting

The manuscript is well-written and generally clear. However, there are a few instances where the language could be further improved for better clarity and precision. Additionally, some sentences or statements could be rephrased to enhance the flow and readability of the manuscript. The figures and tables provided are clear, well-labeled, and effectively support the presentation of the data. However, ensure that all figures and tables are referenced and discussed appropriately in the text. Some minor grammatical errors, typos, and punctuation issues were identified throughout the manuscript. It is recommended to carefully proofread the manuscript to ensure language accuracy.

The abstract effectively introduces the significance of land suitability studies for sustainable agricultural practices. It provides a concise overview of the study conducted in the Tusba District of Van, Turkey, utilizing specific methodologies. However, to enhance the clarity and completeness of the abstract, a few minor comments are offered.

1. The abstract presents the creation of a Soil Quality Index map and its classifications, but consider mentioning some key findings or insights obtained from the study. This could provide readers with a glimpse into the outcomes of the research.
2. When discussing the positive correlation between soil quality index values and crop yield, consider providing a hint of the magnitude or significance of this correlation if possible.
3. In the closing sentence, emphasize the contribution or novel aspect of the study. For instance, mentioning how this research advances the understanding or application of land suitability studies in a specific context can enhance the abstract's impact.
Introduction

1. In introduction discusses the significance of land suitability assessment, consider explicitly stating the research gap or the specific objective of the current study.
2. The study is conducted in the Alaköy and Atmaca districts of the Tusba-Van region in Turkey, consider briefly elaborating on why this region was chosen. Highlight any unique characteristics or relevance that make these districts particularly suitable for investigation.
3. Explicitly state the primary contribution of the current research. Whether it's the modeling approach, the focus on the specific districts, or the integration of methodologies, highlighting the unique aspect of the study will engage readers and establish the context for the subsequent sections.

Results and discussion:
The "Soil Properties and Maps" section provides a comprehensive overview of the descriptive statistics and properties of the soil samples collected from the study area. The presentation is clear and informative. However, there are a few suggestions

1. Consider structuring the information in a more organized manner, such as presenting the descriptive statistics of each parameter (clay content, silt content, etc.) in a table.

2. Ensure consistency in terminology, such as consistently using "content" or "concentration" throughout the section to describe the proportion of elements or compounds in the soil.

3. Include appropriate units for each measurement to provide a clear understanding of the scale being discussed (e.g., % for content, mol kg-1 for exchangeable ions, mg kg-1 for nutrient concentrations).

4. Establish a clearer connection between the descriptive statistics provided and their relevance to the subsequent land suitability analysis. How do these soil properties contribute to determining the suitability of the study area for wheat-barley cultivation?

5. When discussing the coefficients of variation, skewness, and normal distribution, elaborate on the implications of these findings. How might high variability or specific distribution patterns influence the interpretation of soil suitability for wheat-barley cultivation?

6. If applicable, relate your findings to existing literature or studies. Do your findings align with or diverge from previous research on soil properties and their relevance to crop cultivation?

7. For the RMSE values obtained from the models, discuss what these values imply about the accuracy and reliability of the models in predicting soil parameters. This will provide a clearer understanding of the model performance.



Conclusion section
Effectively summarizes the key findings and implications of the study. To enhance the clarity and impact of the conclusion, a few suggestions.

1. Briefly synthesizing the main findings of the study. This could include summarizing the significance of the land suitability classifications, the importance of ecological balance, and the effectiveness of the modeling approach.

2. Highlight Practical Implications of the research for land and watershed management, planning, and agricultural production. How can the identified land suitability classes guide effective land use? How might farmers benefit from this research in their decision-making?

3. Consider suggesting potential avenues for future research and application. For example, how can these findings be extended to more specific land management plans or integrated with other studies? What areas of agricultural sustainability could benefit from further investigation?

4. If possible, provide a brief quantitative overview of the expected socio-economic impact of the study's findings. This could involve estimated benefits for farmers, potential increases in agricultural production, or other measurable outcomes.

5. Discuss how the study's results might contribute to a broader picture of sustainable agriculture, including potential impacts on soil conservation, resource protection, and efficient land management practices.

Experimental design

no

Validity of the findings

no

Additional comments

no

---

## Round 0.2 · Minor Revisions

Thanks for submitting the revisions. The language of the paper needs attention. Please break down the complex sentences into simple ones and try to convey your ideas with a reader-friendly approach.

**Language Note:** The Academic Editor has identified that the English language must be improved. PeerJ can provide language editing services - please contact us at [email protected] for pricing (be sure to provide your manuscript number and title). Alternatively, you should make your own arrangements to improve the language quality and provide details in your response letter. – PeerJ Staff

Reviewer 2 ·

Basic reporting

Accepted

Experimental design

no

Validity of the findings

no

Additional comments

no

---

## Round 0.3 · Minor Revisions

The paper is improved after revisions. Still, there are many grammatical errors. For instance, overuse of punctuation is observed in some places. Moreover, complex sentences need to be broken into simple sentences to be reader-friendly.

**Language Note:** The Academic Editor has identified that the English language must be improved. PeerJ can provide language editing services - please contact us at [email protected] for pricing (be sure to provide your manuscript number and title). Alternatively, you should make your own arrangements to improve the language quality and provide details in your response letter. – PeerJ Staff

---

## Round 0.4 · Minor Revisions

I recommend editing of language. There are still many grammar mistakes. I have mentioned some examples. For instance:

In 36, Keywords instead of keyword
In lines 46, 56, 246, and 421, the double full stop is used.
The comma should be used:
In line 21 i.e., Process method(,) and Standard Scoring FunctionIn line 58, planning(,) and consolidation by determining
In line 58, planning(,) and consolidation by determining
In line 62, topography and climate(,) and quantitative
In line 143, 50% at 13 dS m-(,) and 100% at 20 dS m-1
In line 251, Cu(,) and Mn contents
In line 252, .65 mg kg-1(,) and 4.78-
In line 253, 11.85, 0.61, 1.34(,) and 33.95
In line 256, (,)and to make the criteria
In line 265, bulk density(,) and pH
In line 271, EC, OM, CaCO3(,) and ESP
In line 287, chemical(,) and productivity
In line 290, in the Figure 3.
In line 299, productivity(,) and main parameters
In line 315, input use (,) and soil quality

Other mistakes:

In 76, Considering (the) current situation
In 81, (the) Wheat-barley
In 155, given (in) Table 2.
In 169, (a or the) negative scoring function
235, space 10.53and
There are many more.

**Language Note:** The Academic Editor has identified that the English language must be improved. PeerJ can provide language editing services - please contact us at [email protected] for pricing (be sure to provide your manuscript number and title). Alternatively, you should make your own arrangements to improve the language quality and provide details in your response letter. – PeerJ Staff

---

## Round 0.5 · accepted · Accept

Thanks for incorporating the suggested changes. The paper is improved and accepted for publication.